# Cobalt Protoporphyrin Blocks EqHV-8 Infection via IFN-α/β Production

**DOI:** 10.3390/ani13172690

**Published:** 2023-08-22

**Authors:** Liangliang Li, Xinyao Hu, Shuwen Li, Ying Li, Shengmiao Zhao, Fengzhen Shen, Changfa Wang, Yubao Li, Tongtong Wang

**Affiliations:** College of Agronomy, Liaocheng University, Liaocheng 252000, China

**Keywords:** CoPP, EqHV-8, HO-1, antiviral therapeutics

## Abstract

**Simple Summary:**

Equid alphaherpesvirus type 8 (EqHV-8) infection presents equids with severe respiratory disease, abortions, and neurological syndromes. No vaccines and therapeutic molecules have been reported for EqHV-8 control. In the present study, cobalt protoporphyrin (CoPP) possesses antiviral activity against EqHV-8 via HO-1 (heme oxygenase-1) mediated type I interferon (IFN) response; it will be a novel potential molecule to develop an effective therapeutic drug for EqHV-8 prevention.

**Abstract:**

Equid alphaherpesvirus type 8 (EqHV-8) is the causative agent of severe respiratory disease, abortions, and neurological syndromes in equines and has resulted in huge economic losses to the donkey industry. Currently, there exist no therapeutic molecules for controlling EqHV-8 infection. We evaluated the potential antiviral activity of cobalt protoporphyrin (CoPP) against EqHV-8 infection. Our results demonstrated that CoPP inhibited EqHV-8 infection in susceptible cells and mouse models. Furthermore, CoPP blocked the replication of EqHV-8 via HO-1 (heme oxygenase-1) mediated type I interferon (IFN) response. In conclusion, our data suggested that CoPP could serve as a novel potential molecule to develop an effective therapeutic strategy for EqHV-8 prevention and control.

## 1. Introduction

Equid alphaherpesvirus 8 (EqHV-8) is a double-stranded enveloped DNA virus that belongs to the *Alphaherpesvirinae* subfamily [1]. It has been reported to cause severe respiratory diseases, abortions, and neurological disorders in equines [2]. Recently, numerous EqHV-8 field strains have been isolated from horses or donkeys in several countries and have caused enormous economic losses to the equine and donkey industry worldwide [3,4]. Unfortunately, no effective drug is available to control EqHV-8 infection, indicating the urgency to develop more drugs to control and prevent EqHV-8 infections.

HO-1 (heme oxygenase-1), encoded by the *HMOX1* gene, a rate-limiting enzyme implicated in heme catabolism, has been reported to exert anti-inflammatory, anti-apoptotic, antioxidative, and antiviral effects in the host [5,6,7,8,9,10]. Recent studies have reported a broad-spectrum antiviral effect of HO-1 against numerous viruses, including human immunodeficiency virus (HIV), human respiratory syncytial virus (hRSV), hepatitis B virus (HBV), hepatitis C virus (HCV), Ebola virus (EBOV), bovine viral diarrhea virus (BVD), porcine reproductive and respiratory syndrome virus (PRRSV), and hepatitis A virus (HAV) [11,12,13,14,15,16,17]. Type I IFNs, known as viral IFNs, are induced by virus infection; most types of virally infected cells are capable of synthesizing IFN-α/β in cell culture, and they play a critical role in antivirus infections [18,19]. For example, dengue virus (DENV) replication is sensitive to IFN in both in vitro and in vivo studies [20,21]. The relationship between HO-1 and innate immunity has been widely studied. For example, GV1001 exerts an anti-HBV activity via HO-1-mediated production of type I interferon (IFN) [22]. During pathogen invasion, HO-1 regulates the activation of molecules involved in innate immunity such as interleukin (IL)-6, tumor necrosis factor (TNF), and nitric oxide (NO) [23]. To our knowledge, anti-EqHV-8 activity and the potent mechanism of CoPP have not been reported until now.

A previous study has reported that COPP, a potent agonist of HO-1, possesses antiviral effects. In the present study, we investigated the activity of CoPP against EqHV-8 and the potential antiviral mechanisms of anti-EqHV-8. We demonstrated that CoPP exerts a strong anti-EqHV-8 activity both in vitro and in vivo. Furthermore, the anti-EqHV-8 effect of CoPP depends on HO-1-mediated type I IFN production, which could provide clues for developing new drugs against EqHV-8 infection.

## 2. Materials and Methods

### 2.1. Cells, Viruses, Antibodies, and Chemicals

MH-S (a murine alveolar macrophage cell) and MDBK (Madin–Darby bovine kidney) cells were purchased from the China Center for Type Culture Collection (CCTCC, Wuhan, China) and maintained in 10% fetal bovine serum (FBS) Dulbecco’s minimal essential medium (DMEM) at 37 °C and 5% CO_2_. The EqHV-8 SDLC66 strain (GenBank: MW816102.1) was proliferated in MDBK cells. The mouse anti-EqHV-8-positive serum, derived from mice by artificial infection EqHV-8, was prepared in our laboratory, and all experimental protocols were approved by the Liaocheng University Animal Care and Use Committee (permit number: LC2021-05). Anti-OAS1 (2′-5′-oligoadenylate synthetases 1) (catalog number ab272492), anti-PKR (Protein kinase R) antibodies (catalog number ab184257), and anti-HO-1 mAb (catalog number ab13243) were obtained from Abcam, and Cy3 AffiniPure goat anti-mouse IgG (H+L) (catalog number 115165044) and horseradish peroxidase (HRP)-labeled goat anti-mouse IgG (H+L) (catalog number 115035020) were purchased from Jackson. CoPP was purchased from Sigma-Aldrich (St. Louis, MO, USA) and dissolved in NaOH (0.2 M).

The MDBK cells were seeded into 96-well plates for 24 h before virus infection, following which the viral supernatant was serially diluted 10-fold in eight replicates with 100 μL per well. The tissue culture infectious dose 50 (TCID_50_) was calculated after 5 days post-infection (dpi). The production of virus progeny was determined in MDBK cells using the Reed–Muench method, and the data were analyzed using GraphPad Prism 8.0.

### 2.2. CoPP Cytotoxicity

MH-S and MDBK cells were seeded into 96-well plates (1 × 10^4^/well). After 24 h, varying concentrations (5, 10, 25, 50, and 100 μM) of CoPP were added to the wells, and the cells were incubated for 24 h. Afterward, the CCK-8 (Cell Counting Kit-8) reagent (10 μL/well) was added, and the cells were incubated for 2 h. The number of viable cells was determined by reading the absorbance at 450 nm as measured by the EpochTM Microplate spectrophotometer (BioTek, Winooski, VT, USA), and these data were analyzed using GraphPad Prism 8.0.

### 2.3. Inhibition Assay of CoPP In Vitro

To investigate the potential antiviral effect of CoPP against EqHV-8 infection in vitro, MH-S and MDBK cells were pre-treated with different concentrations of CoPP (0, 5, 10, 25, 50, and 100 μM) for 12 h, followed by infection with EqHV-8 SDLC66 (0.1 MOI). The cell samples and cellular supernatant were harvested at 24 h post-infection (hpi) and analyzed by Western blotting and qPCR.

### 2.4. Effects of CoPP on EqHV-8 Infection

To determine the stage of the EqHV-8 life cycle affected by CoPP, MH-S cells were seeded into 12-well plates and termed as pre-treated, co-treated, and post-treated with CoPP (100 µM) relative to the EqHV-8 (0.1 MOI) inoculation groups. After 24 h, the cells were collected to examine the expression of the gD protein, and the production of progeny viruses was measured by Western blotting using anti-EqHV-8 gD polyclonal antibody and qPCR with EqHV-8 186 primers as described above.

### 2.5. The Change of IFN Response Treated with CoPP

The siRNAs targeting the *HO-1* gene were synthesized by Ribo Biotechnology Co., Ltd. (Guangdong, China), and the siRNAs sequences are shown in Table 1. The effects of siRNAs were identified by Western blotting. MH-S cells were transfected with si-HO-1 (100 nM) or siNC (negative-control) (100 nM), followed by treatment with CoPP (100 μM) for 12 h. Subsequently, the cells were infected with EqHV-8 SDLC66 with 0.1 MOI. All cells were collected and RNA was extracted to evaluate the expression of *IFNα, IFNβ, HO-1*, *gD*, and interferon-stimulated genes (ISGs) (*PKR* and *OAS1*) by qPCR; simultaneously, HO-1, gD, PKR, and OAS1 were detected by Western blotting, and then cells were treated with si-HO-1 or siNC; the expression of HO-1, gD, PKR, and OAS1 proteins was also detected by Western blotting. The data were analyzed by GraphPad Prism 8.0 and the ChemiDoc XRS imaging system.

### 2.6. Animal Experiments

Twelve specific pathogen-free, 6-week-old male BALB/c mice were purchased from the Pengyue Experimental Animal Breeding Co., Ltd. (Jinan, China) and randomly divided into four groups (*n* = 3 mice/group). Mice in group 1 were inoculated with the DMEM medium as the Mock group. Mice in group 2 were inoculated intraperitoneally with 100 μL of DMEM, and those in group 3 were inoculated intraperitoneally with 100 μL of NaOH solution (0.2 M). Mice in group 4 were inoculated intraperitoneally with 100 μL of CoPP (20 μM/kg). Mice in group 2, group 3, and group 4 were challenged intranasally with 1 × 10^5^ PFU/mice of EqHV-8. EqHV-8 or DMEM incubation in mice was performed under deep anesthesia with Zoletil 50 (Virbac, Nice, France). Mice in each group were housed separately to prevent cross-infection. The clinical symptoms of mice were monitored daily. Finally, the mice were euthanized at 8 dpi via cervical dislocation for subsequent experiments on pathological changes and virus replication in the lungs.

#### 2.6.1. Histopathology Evaluation

The mice lungs of different groups at 8 dpi were collected for hematoxylin and eosin (HE) staining for evaluating histopathological changes as previously described [24]. Briefly, lungs were fixed in 10% formalin solution, underwent dehydration by alcohol, were transparentized in xylene, and then embedded in paraffin wax, sliced in a microtome (Leica, Nussloch, Germany) to 4 µm, affixed onto slides, followed by deparaffinization, clearance, diluted alcohols (95%, 75%), and hematoxylin solution stains, were differentiated, using 95% or 100% alcohol directly after the eosin stain, and, after cleared, placed on a coverslip to be observed by light microscopy.

#### 2.6.2. Virus Replication in Tissues

Lungs (0.1 g) from different groups mixed with PBS (1 mL) were crushed, homogenized, and then frozen and thawed 3 times. After that, the supernatant was collected to extract viral genomic DNA using a Viral DNA Kit (Omega Bio-Tek, Inc., Norcross, GA, USA) according to the manufacturer’s protocol. To determine the EqHV-8 replication in the lungs, a qPCR assay was used to detect EqHV-8 based on the above methods. Finally, viral loads were calculated in the lungs by absolute quantification.

### 2.7. Gene Transcription and Immunoblot Analysis

#### 2.7.1. Real-Time Quantitative PCR (qPCR)

All cell samples were collected and analyzed using a step-one plus real-time polymerase chain reaction (PCR) system as previously described [24]. Briefly, the total RNA was extracted by TRIzol (Invitrogen, Carlsbad, CA, USA) according to the manufacturer’s instructions, and reverse transcription PCR was performed using the PrimeScript™ RT Master Mix kit (Takara, Japan). The mRNA transcription levels of HO-1, glycoprotein D protein(gD), IFN-α, IFN-β, PKR, and OAS1 were normalized against those of glyceraldehyde 3-phosphate dehydrogenase (GAPDH) using the 2^−ΔΔCT^ threshold cycle (CT) method, and the relative fold changes were subsequently calculated. All primers are listed in Table 1.

To determine the EqHV-8 genome DNA copy number, the absolute quantification was performed using the pMD18-T-ORF72 as the template, containing 186 bp of EqHV-8 ORF72. It served as a standard sample to calculate the EqHV-8 genome DNA copies. The qPCR reaction was performed at 95 °C for 4 min, followed by 40 cycles at 94 °C for 30 s, 60 °C for 30 s, and 72 °C for 20 s.

#### 2.7.2. Western Blotting

The cells were collected and lysed in the radioimmunoprecipitation (RIPA) lysis buffer and mixed to boil for 10 min in with 5× sample loading buffer. Next, the proteins were loaded onto 12% sodium dodecyl sulfate–polyacrylamide gel electrophoresis (SDS–PAGE) gels with equal amounts and transferred onto polyvinylidene fluoride (PVDF) membranes as described previously [25]. The polyvinylidene fluoride (PVDF) membranes were blocked with 5% bovine serum albumin (BSA) and incubated with anti-HO-1 mAb, anti-α-tubulin mAb, or anti-EqHV-8 gD polyclonal antibody, PKR, and OAS1 antibodies. HRP-conjugated goat anti-mouse or goat anti-rabbit IgG was used as the secondary antibody. Finally, the protein band signals were detected using an enhanced chemiluminescent (ECL) kit (Bio-Rad, San Francisco, CA, USA). The images were analyzed using the ChemiDoc XRS imaging system (Bio-Rad).

### 2.8. Statistical Analysis

Statistical analysis was performed using GraphPad Prism software. Differences among the groups were analyzed by unpaired Student’s t-test. Significance is indicated as follows: *, *p* < 0.05; **, *p* < 0.01; and ***, *p* < 0.001.

## 3. Results

### 3.1. Chemical Structure and CoPP Cytotoxicity

The structure of CoPP is illustrated in Figure 1A. The cytotoxicity of CoPP in MH-S and MDBK cells was determined using the CCK-8 kit. The results showed that CoPP did not exert a cytotoxic effect at concentrations lower than 100 μM in these cells (Figure 1B).

### 3.2. Inhibition Assay of CoPP In Vitro

The results demonstrated that CoPP decreased the expression of gD protein in MH-S cells in a dose-dependent manner (Figure 2A,B). Similar to the results of the protein analysis, the production of progeny viruses was reduced in the CoPP-treated group compared with the control group (Figure 2C). Similar results were observed in MDBK cells (Figure 2D–F).

### 3.3. Effects of CoPP on EqHV-8 Infection

To determine if the replication cycle of EqHV-8 is affected by CoPP, we performed a time-of-addition experiment as shown in Figure 3A. M1 stands for the EqHV-8 infection group, which served as a positive control (without CoPP treatment). M2 stands for the CoPP pre-treated group. M3 stands for the CoPP and EqHV-8 co-treated group, whereas M4 stands for the CoPP post-treated group. The MH-S cells were harvested, and the supernatant was collected to measure the expression of gD protein and assess the production of progeny viruses at 24 hpi. Both the expression of gD protein and the number of copies of the virus significantly decreased in the M2, M3, and M4 groups, as compared with the M1 control group (Figure 3B,C), indicating that CoPP inhibits EqHV-8 infection at multiple stages of the virus life cycle.

### 3.4. The Change of IFN Response Treated with CoPP

HO-1 has been reported to function as a critical mediator of innate immunity by regulating the production of IFN [26,27,28]. Firstly, to evaluate whether HO-1 activity is a critical factor for CoPP against EqHV-8 infection, MH-S cells were pre-treated with CoPP or si-HO-1, followed by infection with EqHV-8. Next, the expression of *HO-1* and *gD* genes was analyzed in these cells. The results showed that CoPP reduced the expression of the gD gene by inducing HO-1, which was reversed by treatment with si-HO-1 at both transcription and protein levels (Figure 4A,B). To further determine whether the anti-EqHV-8 activity depends on HO-1-medicated IFN response, we pre-treated the MH-S cells with different concentrations of CoPP, followed by infection with EqHV-8. Next, the expression of IFNα/β and ISGs (PKR, OAS1) was analyzed. The expression of IFN α/β (Figure 4C,D) and ISGs (Figure 4E) increased markedly following CoPP treatment. Treatment with si-HO-1 reversed the anti-EqHV-8 activity of CoPP via HO-1 knockdown, and the expression of PKR and OAS1 proteins was reduced (Figure 4F). These data indicated that the anti-EqHV-8 activity of CoPP largely depends on HO-1-mediated IFN production.

### 3.5. CoPP Decreased EqHV-8 Infection in Mice Model

The BALB/c mice models can be used to study virus replication and virulence of EqHV-8 as described previously [29]. To confirm the antiviral effect of CoPP against EqHV-8 infection in vivo, we assessed the viral load, pathological lesions, and cytokine levels in the lungs of CoPP-treated EqHV-8-infected mice and compared them with the EqHV-8-infected group. Results demonstrated that one mouse died at 6 dpi, 7 dpi, and 8 dpi, respectively, in the EqHV-8-infected group or NaOH-treated groups; however, only one mouse died at 8 dpi in the CoPP-treated group (Figure 5A). CoPP alleviated the clinical symptoms of EqHV-8 infection and reduced the mortality rate by 66.7%. The number of viral DNA copies in CoPP-treated lungs was significantly lower than that in the EqHV-8-infected group and NaOH-treated groups (Figure 5B). To further characterize the protective effects of CoPP on EqHV-8-caused lung damage, we examined the histopathological changes. As expected, compared with the EqHV-8 infected group, CoPP reduced EqHV-8-induced interstitial pneumonia (characterized by thicker alveolus walls and inflammatory cell infiltration) in the lung tissues (Figure 5C). In addition, the NaOH-treated group showed no discrepancy with the EqHV-8 infected group. Mice in the CoPP-treated group displayed significantly enhanced expression of HO-1 and IFN-α/β compared with the EqHV-8 infected group, whereas no significant discrepancy was observed in the NaOH-treated mice (Figure 5D). These data suggested that the anti-EqHV-8 activity of CoPP largely depends on HO-1 mediated IFNα/β generation (Figure 6).

## 4. Discussion

EqHV-8 has emerged as an important and prevalent viral pathogen of donkeys. The infection has resulted in huge economic losses and is a major threat to the donkey industry worldwide. EqHV-8 was first reported in the nasal cavity of latently infected donkeys in Australia in 1988 [30]. It was isolated from a horse with a fever and runny nose in China [3]. Subsequently, EqHV-8 was reported in a donkey from Israel. There has been a recent increase in the large-scale breeding of donkeys in China. EqHV-8 infection is characterized by abortion and respiratory diseases that seriously hinder the economic growth of the donkey industry. Our group previously reported that EqHV-8 causes abortion and respiratory diseases in donkeys [4]. However, the number of effective drugs available against EqHV-8 infection is limited. In the present study, we demonstrated that CoPP exerts anti- EqHV-8 activity via HO-1-mediated production of type I IFNs and ISGs (Figure 6).

CoPP is an analog of the heme group and contains a central iron moiety. It is a well-known inducer of antioxidants and HO-1 [31]. Recently, increasing evidence has demonstrated that CoPP protects host cells from viral damage by upregulating the expression of HO-1. For instance, Ma et al. reported that CoPP inhibits the replication of the influenza virus via IRF3-mediated generation of IFN-α/β [28]. Similar antiviral effects of CoPP were observed against the spring viremia of carp virus (SVCV) through ROS/Nrf2/HO-1 axis; CoPP inducing the HO-1 signal pathway was a promising strategy for treating Duck Tembusu virus (DTMUV) infection, hepatitis A virus (HAV), and pseudorabies virus (PRV), etc., as shown in recent studies [17,32,33,34]. Except for the Zika virus (ZIKV), induction of HO-1 by CoPP to limit ZIKV infection may be ineffective as a therapeutic strategy because ZIKV was able to downregulate HO-1 expression [35]. In the present study, CoPP significantly inhibited the EqHV-8 infection in MH-S and MDBK cells in a dose-dependent manner (Figure 2). To further explore the antiviral mechanisms of CoPP, we first determined which stages of the viral replication cycle were affected by CoPP. Our data suggested that CoPP interfered with multiple replication processes of EqHV-8 (Figure 3). In addition, our results suggested that CoPP inhibited EqHV-8 replication by upregulating the production of type I IFNs and ISGs (PKR, OAS1) mediated by HO-1 in vitro (Figure 4), Interestingly, our results confirmed that CoPP improved clinical symptoms and reduced the replication of EqHV-8 in the lungs of mice model (Figure 5A–C). Meanwhile, The HO-1 and type I IFNs expressions were also confirmed using the mice model (Figure 5D).

According to previous studies, mouse model was the main animal model for researching the effects of CoPP on viral infections, such as hRSV, DENV, HBV [36], and CoPP, which were applied with different concentrations as antiviral molecules. For example, Tseng et al. used 50 mg/kg CoPP to treat mice, followed by infecting DENV, which suggested a significant delay in the onset of disease and mortality, and decreased virus load in the infected mice’s brains [37]. Protzer et al. used 10 mg/kg CoPP to indicate it protected mice from immune-mediated hepatitis associated with apoptotic liver damage induced by HBV infection [38]. Espinoza et al. used 7.6 μΜ/kg (5 mg/kg) CoPP to protect mice from hRSV damage; HO-1 induction also decreased virus replication and lung inflammation [12]. In our study, 20 uM/kg (30.5 mg/kg) CoPP was used to confirm the inhibition effect of EqHV-8 replication in mice model. The above examples suggested that the concentration of CoPP at 5–50 mg/kg is safe for mice growth and physiological function. However, other species that applied CoPP in antivirus research have not been reported until now, and in the future, we will attempt to verify the potential effectiveness of CoPP against EqHV-8 in equids, and a more extensive range of CoPP concentrations will be investigated to determine optimal dosages. It will provide a novel the potential application and effectiveness of CoPP as an antiviral molecule, and also help identify any similarities or differences in the mechanisms of action observed across different viruses and species.

## 5. Conclusions

In the present study, our data demonstrated that CoPP suppresses the replication of EqHV-8 in vitro and in vivo via HO-1-mediated production of IFN-α/β, indicating that CoPP could be a potential novel drug for EqHV-8 control.

## Figures and Tables

**Figure 1 animals-13-02690-f001:**
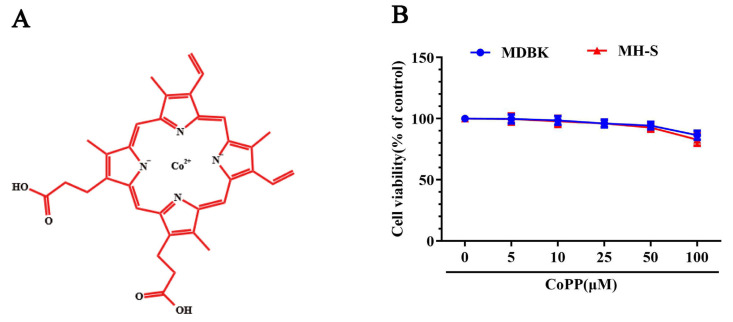
Chemical structure and cytotoxicity of CoPP. (**A**) Chemical structure of CoPP. (**B**) Cytotoxicity of CoPP in MH-S and MDBK cells. Cells were seeded in 96-well cell plates and treated with different concentrations of CoPP (0, 5, 10, 25, 50, and 100 μM) for 24 h. The viability of cells was determined using the CCK-8 assay. These results were confirmed in three independent experiments.

**Figure 2 animals-13-02690-f002:**
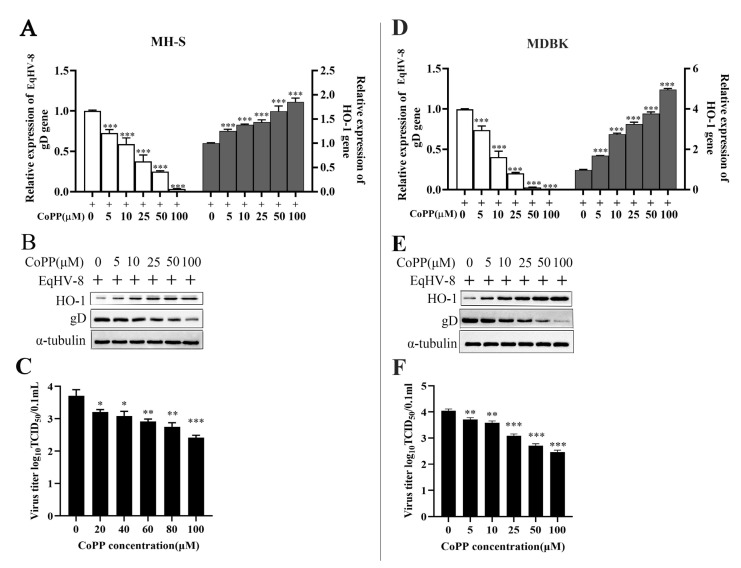
Antiviral activity of CoPP against EqHV-8 in MH-S and MDBK cells. MH-S cells were pre-incubated with different concentrations of CoPP for 12 h and afterward infected with EqHV-8 SDLC66 at 0.1 MOI. The production of gD protein was analyzed by qPCR (**A**) and Western blotting (**B**), and the production of progeny viruses was measured by TCID_50_ (**C**). MDBK cells were treated with CoPP using the same protocol, and the EqHV-8 replication was determined by qPCR (**D**), Western blotting, (**E**), and TCID_50_ (**F**). Cropped blots are displayed; the samples were derived from the same experiment, and gels/blots were processed in parallel. GAPDH served as an internal control, and the data shown are representatives from three independent experiments and subjected to unpaired Student’s *t*-tests. * *p* < 0.05, ** *p* < 0.01, *** *p* < 0.001 (compared with 0 µM CoPP-treated cells).

**Figure 3 animals-13-02690-f003:**
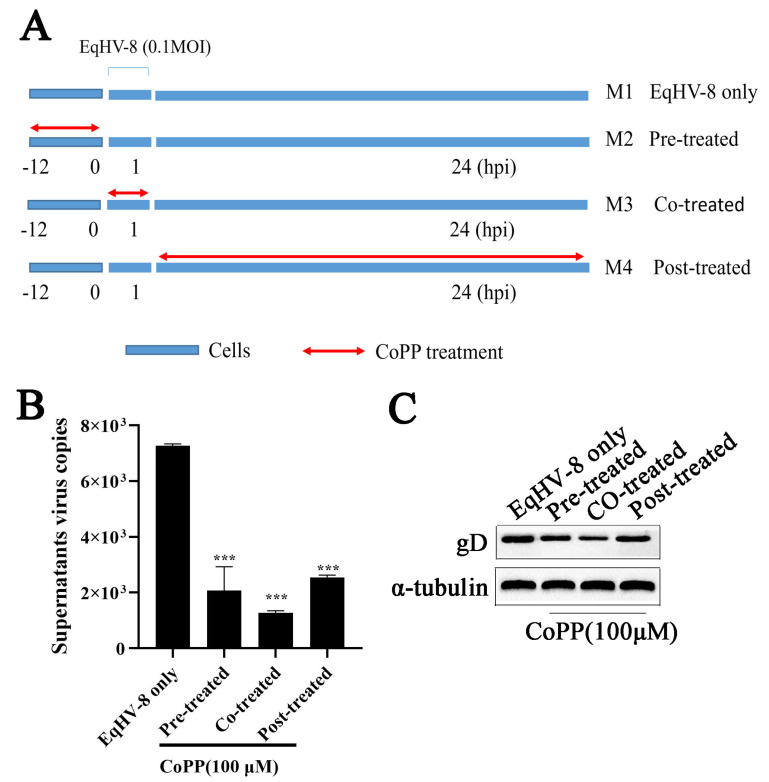
CoPP inhibits EqHV-8 infection at multiple stages of the virus life cycle. MH-S cells were infected with EqHV-8 (0.1 MOI) and treated with CoPP (100 µM) at different times of infection, including pre-treatment, co-treatment, and post-treatment. Schematic diagram of CoPP-treated cells (**A**). The expression of gD protein was determined by qPCR (**B**) and Western blotting (**C**). Cropped blots are displayed; the samples were derived from the same experiment, and gels/blots were processed in parallel. GAPDH served as an internal control. The data shown are representatives from three independent experiments subjected to unpaired Student’s *t*-tests. *** *p* < 0.001 (compared with cells in the EqHV-8 only group).

**Figure 4 animals-13-02690-f004:**
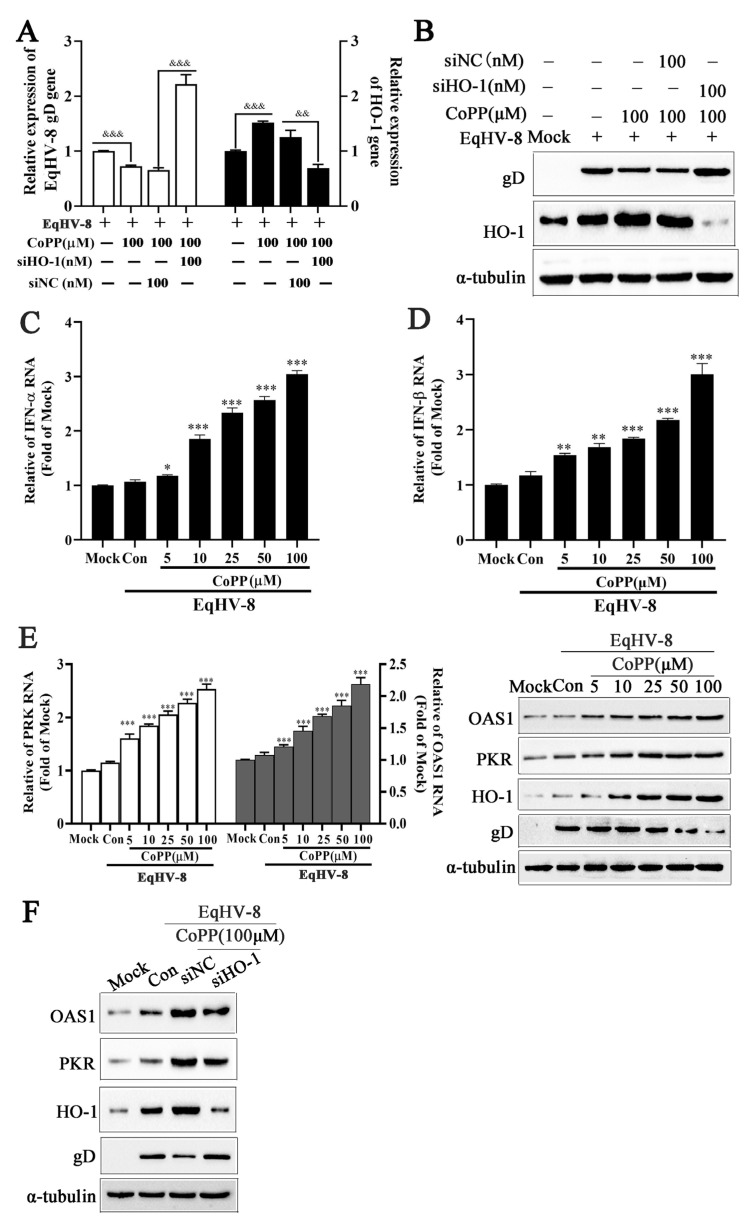
CoPP inhibits EqHV-8 replication by activating HO-1-mediated IFN response. MH-S cells were pre-treated with CoPP (100 μM) or 100 nM siRNAs and afterward infected with EqHV-8 SDLC66 at 0.1 MOI. The expression of HO-1 and gD was determined by qPCR and Western blotting (**A**,**B**). Cropped blots are displayed; the samples were derived from the same experiment, and gels/blots were processed in parallel. The data shown are representatives from three independent experiments subjected to unpaired Student’s *t*-tests. ^&&^
*p* < 0.01, ^&&&^
*p* < 0.001. The effects of CoPP on the expression of IFN-α/β and ISGs (PKR and OAS1). MH-S cells were treated with different concentrations of CoPP and infected with EqHV-8 SDLC66 (0.1 MOI). The mRNA expression of IFN-α/β was determined by qPCR (**C**,**D**). The mRNA and protein levels of PKR and OAS1 were measured by qPCR and Western blotting at 24 hpi (**E**). The relationship of si-HO-1 on the CoPP-induced activation of IFN response and anti-EqHV-8 effect. MH-S cells were transfected with si-HO-1 or si-NC for 12 h and subsequently infected with EqHV-8 SDLC66 (0.1 MOI) in the presence of 100 μM CoPP for 24 h. The protein expression of PKR and OAS1 was determined through Western blotting (**F**). Cropped blots are displayed; the samples were derived from the same experiment, and gels/blots were processed in parallel. The data shown are representatives from three independent experiments subjected to unpaired Student’s *t*-tests. * *p* < 0.05, ** *p* < 0.01, *** *p* < 0.001 versus Mock.

**Figure 5 animals-13-02690-f005:**
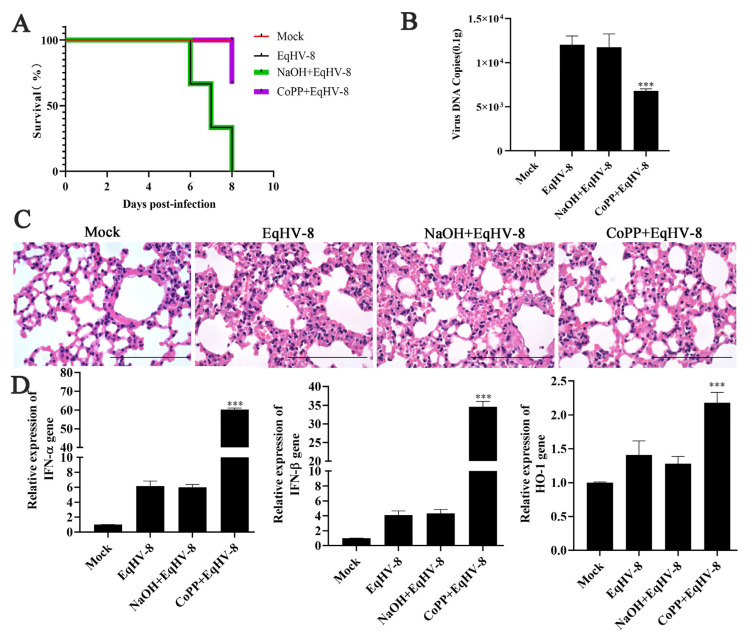
CoPP treatment diminishes the replication and pathogenesis of EqHV-8. Mice were infected by intranasal injection with EqHV-8, and NaOH or CoPP treatment before EqHV-8 infection; DMEM treatment served as the Mock group. (**A**) The survival rate of mice among different groups. (**B**) The numbers of viral genomes in the lungs of mice in the indicated groups were measured by qPCR. (**C**) Representative images of hematoxylin and eosin (H&E) in the lungs derived from mice in the indicated groups. Bar, 100 µm. (**D**) The expression of HO-1 and IFN-α/β in the lung tissues of mice was assessed by qPCR. The data shown are representatives from three independent experiments subjected to unpaired Student’s *t*-tests. *** *p* < 0.001 (compared with the EqHV-8 infected group).

**Figure 6 animals-13-02690-f006:**
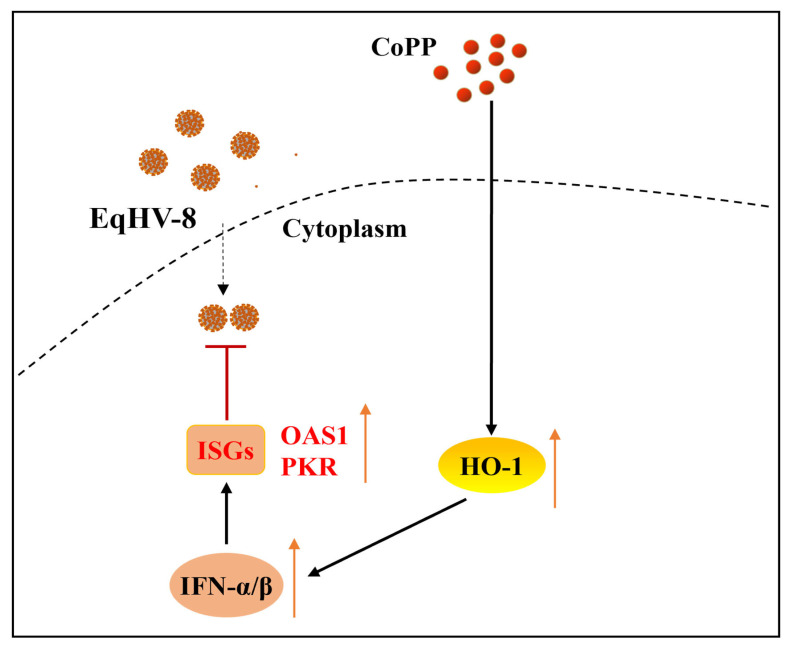
Schematic showing that CoPP reduces EqHV-8 infection. CoPP increased the expression and activity of the *HO-1* gene, leading to host cellular type I IFN response and ISG (OAS1 and PKR) expression, and, subsequently, decreased EqHV-8 replication.

**Table 1 animals-13-02690-t001:** The primers in this study.

Primers	Primer Sequences (5′-3′)
*HO-1*-F	AGTTCATGAAGAACTTTCA
*HO-1*-R	TACCAGAAGGCCATGTCC
EqHV-8 186-F	CCCACGTGTGCAACGCCTAT
EqHV-8 186-R	ATACAGTCCCGAGGCAGAGT
EqHV-8-gD-F	GATGCCAAACCGAATCAGCC
EqHV-8-gD-R	TAGGCGAGTCAAGCCGTTTT
*IFN-α*-F	TACTCAGCAGACCTTGAACCT
*IFN-α*-R	CAGTATTGGCAGCAAGTTGAC
*IFN-β*-F	AGCTCCAAGAAAGGACGAACAT
*IFN-β*-R	GCCCTGTAGGTGAGGTTGATCT
*OAS1*-F	GGAGGCGGTTGGCTGAAGAGG
*OAS1*-R	GAACCACCGTCGGCACATCC
*PKR*-F	CGTTTCTTGCCTCCTGCTTTG
*PKR*-R	GGGACCTCCACATGACAGAAG
*GAPDH*-F	CCTTCCGTGTCCCTACTGCCAAC
*GAPDH*-R	GACGCCTGCTTCACCACCTTCT

## Data Availability

The original contributions presented in this study are included in the article. Further inquiries can be directed to the corresponding authors.

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
