# Peer review of "Cobalt Protoporphyrin Blocks EqHV-8 Infection via IFN-α/β Production"

_animals, 2023, doi:10.3390/ani13172690_

Round 1
Reviewer 1 Report
Cobalt protoporphyrin blocks EHV-8 infection via IFN-α/β production is a paper that investigate the effect of Cobalt protoporphyrin on the replication levels of EHV-8. This virus is assuming a relevant importance in equine medicine. The research is quite well-prepared, but I think the paper need to be revised, as some parts are difficult to follow, especially M&M and some sentences in R, should be moved in M&M. Discussion is lacking completely, no mention is made of results regarding equine practice. I am not convinced of the statistics you used: why don't you investigate correlation among level of CoPP and virus?
As following a list of observations:
Material and Methods must be revised. The titles of the subsections must be the same of results to be clearly connected. The experiment and its phases must be better described with the rationale of each part (that is written in the results instead). Separate the methods employed from the experimental phases.
L55 The anti-EHV-8-positive serum was prepared in our laboratory, How? Which species did you inoculated? Do you have approval of ethic committee? Please insert Number od Approval.
Please all the acronyms must be explained (OAS, CCK-8 etc).
mock group: is not simpler use control group?
L 224 As expected, compared with the EHV-8 infected group, CoPP reduced EHV-8-induced interstitial pneumonia in the lung tissues. How do you demonstrate this sentence?
<Figure 5. Show the survival rate, but no mention in the text of the fact the mice died apart from the culling.
Author Response
Reviewer #1:
Major points:
Cobalt protoporphyrin blocks EHV-8 infection via IFN-α/β production is a paper that investigate the effect of Cobalt protoporphyrin on the replication levels of EHV-8. This virus is assuming a relevant importance in equine medicine. The research is quite well-prepared, but I think the paper need to be revised, as some parts are difficult to follow, especially M&M and some sentences in R, should be moved in M&M. Discussion is lacking completely, no mention is made of results regarding equine practice. I am not convinced of the statistics you used: why don't you investigate correlation among level of CoPP and virus?
As following a list of observations:
- Material and Methods must be revised. The titles of the subsections must be the same of results to be clearly connected. The experiment and its phases must be better described with the rationale of each part (that is written in the results instead). Separate the methods employed from the experimental phases.
Answer:We are very appreciated for your suggestion. We have revised Material and Methods, and confirmed the title same as results in lines 72, 119, 126, in addition, the methods in result have been moved into Material and Methods in lines 81-86 in revised manuscript.
- L55 The anti-EHV-8-positive serum was prepared in our laboratory, How? Which species did you inoculated? Do you have approval of ethic committee? Please insert Number od Approval.
Answer:Thanks for your comments and suggestion. The anti-EHV-8-positive serum was prepared in our laboratory, derived from mice by artificial infection EHV-8, and all experimental protocols were approved by the Liaocheng University Animal Care and Use Committee (permit number: LC2021-05). We have revised this sentence in lines 62-65 in revised manuscript.
- Please all the acronyms must be explained (OAS, CCK-8 etc).
Answer:Thanks for your comments! We have explained all the acronyms in lines 11, 19, 33, 58, 65, 66, 75, 107, 130, 133 in revised manuscript.
- mock group: is not simpler use control group?
Answer:Thanks for your comments and suggestion. In animal experiments, Mock group is only treated with DMEM, it serves as negative control.
- L 224 As expected, compared with the EHV-8 infected group, CoPP reduced EHV-8-induced interstitial pneumonia in the lung tissues. How do you demonstrate this sentence?
Answer:Thanks for your comments! EHV-8-induced interstitial pneumonia is characterized by thicker alveolus walls, inflammatory cell infiltration, published in our previous paper (Hu L, Wang T, Ren H, Liu W, Li Y, Wang C, Li L. Characterizing the Pathogenesis and Immune Response of Equine Herpesvirus 8 Infection in Lung of Mice. Animals (Basel). 2022, 12(19):2495.), we have re-edited this sentence in lines 266-267 in revised manuscript.
- <Figure 5. Show the survival rate, but no mention in the text of the fact the mice died apart from the culling.
Answer:Thanks for your suggestion! We added the described “Results demonstrated that one mouse died at 6 dpi, 7 dpi and 8 dpi, respectively in EqHV8-infected group or NaOH-treated groups, however, only one mouse died at 8 dpi in CoPP-treated group (Fig. 5A)” in lines 258-260 in revised manuscript.
Reviewer 2 Report
The study conducted by Li et al., titled "Cobalt Protoporphyrin Blocks EHV-8 Infection via IFN-α/β Production," sought to assess the potential antiviral effects of cobalt protoporphyrin (CoPP) against equid herpesvirus type 8 (EHV-8) infection, a virus known to cause severe respiratory disease, abortions, and neurological syndromes in equids. Currently, there are no reported vaccines or therapeutic agents available for the control of EHV-8. The researchers' findings demonstrated that CoPP effectively inhibited EHV-8 infection in both susceptible cells and a mouse model. Moreover, CoPP was shown to hinder the replication of EHV-8 through the HO-1-mediated type I interferon (IFN) response. In conclusion, the authors suggested that CoPP holds promise as a novel and potential candidate for developing an effective therapeutic strategy to prevent and control EHV-8. Overall, the study is of significant importance and can be accepted with only minor revisions.
Minor revisions
1) Page 1, line 24: Please update the virus nomenclature according to ICTV guidelines: Varicellovirus equidalphaherpesvirus 8 (EqHV-8). The information is available at: https://ictv.global/report/chapter/orthoherpesviridae/orthoherpesviridae/varicellovirus
2) Page 2, line 52: Please provide a description for the acronyms of both cell lineages: MH-S and MDBK.
3) Page 2, line 63: Similarly, please provide a description for the CCK-8 reagent.
4) Page 4, line 140: In Figure 1B, the red line represents the average percentage of cell viability for MH-S cells exposed to various concentrations of CoPP (0, 5, 10, 25, 50, and 100 μM) over a duration of 24 hours. The absence of standard deviation plotted in the figure is indeed a valid point of consideration. To enhance the data representation and reinforce the statistical significance, it is recommended to include error bars in future figures, which will add clarity and credibility to the results.
5) In Figure 5A, the choice of colors and line types for the groups makes it difficult to identify and separate them. The authors should improve the presentation of the figure to make it clearer and facilitate the visualization of the distinct groups.
6) Authors should include a paragraph discussing the limitations of the study. Several constraints were identified, emphasizing the necessity to evaluate CoPP's antiviral potential in clinically relevant animal models, specifically horses or mules. Additionally, a more extensive range of CoPP concentrations should be investigated to determine optimal dosages. Understanding the potential development of viral resistance is crucial to assess the long-term efficacy of CoPP as an antiviral treatment. Moreover, comprehensive assessments of the compound's safety and toxicity in non-infected cells and animals are essential prerequisites to ensure its viability as a safe and effective antiviral therapeutic strategy.
7) The authors need for a more comprehensive analysis and comparison of the study results with other research that has evaluated the antiviral activity of CoPP against different species and viruses. The authors could consider discussing and comparing their findings with relevant studies that have investigated the effects of CoPP on other viral infections in various animal models. This comparison would provide a broader context for understanding the potential application and effectiveness of CoPP as an antiviral agent. It would also help identify any similarities or differences in the mechanisms of action observed across different viruses and species. Furthermore, the authors might address any existing gaps or discrepancies between their study results and previous research on CoPP's antiviral properties. By acknowledging these discrepancies, they can provide insights into potential factors influencing the variability of CoPP's antiviral effects and propose future directions for more in-depth investigations. By including a more comprehensive discussion and comparison with related studies, the authors can strengthen the significance of their findings and offer a clearer perspective on the potential implications of CoPP as a therapeutic agent for viral infections.
8) Below is a list of studies that the authors can use to better discuss the work:
8.1) Ma LL, Zhang P, Wang HQ, Li YF, Hu J, Jiang JD, Li YH. heme oxygenase-1 agonist CoPP suppresses influenza virus replication through IRF3-mediated generation of IFN-α/β. Virology. 2019 Feb;528:80-88. doi: 10.1016/j.virol.2018.11.016.
8.2) Toro A, Ruiz MS, Lage-Vickers S, Sanchis P, Sabater A, Pascual G, Seniuk R, Cascardo F, Ledesma-Bazan S, Vilicich F, Vazquez E, Gueron G. A Journey into the Clinical Relevance of Heme Oxygenase 1 for Human Inflammatory Disease and Viral Clearance: Why Does It Matter on the COVID-19 Scene? Antioxidants (Basel). 2022 Jan 29;11(2):276. doi: 10.3390/antiox11020276.
8.3) El Kalamouni C, Frumence E, Bos S, Turpin J, Nativel B, Harrabi W, Wilkinson DA, Meilhac O, Gadea G, Desprès P, Krejbich-Trotot P, Viranaïcken W. Subversion of the Heme Oxygenase-1 Antiviral Activity by Zika Virus. Viruses. 2018 Dec 20;11(1):2. doi: 10.3390/v11010002. PMID: 30577437; PMCID: PMC6356520.
8.4) Zhong M, Wang H, Ma L, Yan H, Wu S, Gu Z, Li Y. DMO-CAP inhibits influenza virus replication by activating heme oxygenase-1-mediated IFN response. Virol J. 2019 Feb 20;16(1):21. doi: 10.1186/s12985-019-1125-9. PMID: 30786886; PMCID: PMC6381609.
8.5) Liu H, Li C, He W, Chen J, Yang G, Chen L, Chang H. Free ISG15 inhibits Pseudorabies virus infection by positively regulating type I IFN signaling. PLoS Pathog. 2022 Oct 31;18(10):e1010921. doi: 10.1371/journal.ppat.1010921. PMID: 36315588; PMCID: PMC9648840.
Author Response
Reviewer #2:
Major points:
The study conducted by Li et al., titled "Cobalt Protoporphyrin Blocks EHV-8 Infection via IFN-α/β Production," sought to assess the potential antiviral effects of cobalt protoporphyrin (CoPP) against equid herpesvirus type 8 (EHV-8) infection, a virus known to cause severe respiratory disease, abortions, and neurological syndromes in equids. Currently, there are no reported vaccines or therapeutic agents available for the control of EHV-8. The researchers' findings demonstrated that CoPP effectively inhibited EHV-8 infection in both susceptible cells and a mouse model. Moreover, CoPP was shown to hinder the replication of EHV-8 through the HO-1-mediated type I interferon (IFN) response. In conclusion, the authors suggested that CoPP holds promise as a novel and potential candidate for developing an effective therapeutic strategy to prevent and control EHV-8. Overall, the study is of significant importance and can be accepted with only minor revisions.
Minor revisions
- Page 1, line 24: Please update the virus nomenclature according to ICTV guidelines: Varicellovirus equidalphaherpesvirus 8 (EqHV-8). The information is available at: https://ictv.global/report/chapter/orthoherpesviridae/orthoherpesviridae/varicellovirus
Answer:We are very appreciated for your suggestion. We have updated nomenclature of EHV-8 to “equid alphaherpesvirus 8 (EqHV8)”, and replaced in full-text and Figures.
- Page 2, line 52: Please provide a description for the acronyms of both cell lineages: MH-S and MDBK.
Answer:Thanks for your suggestion! We have described MH-S (a murine alveolar macrophage cell line) and MDBK (Madin-Darby Bovine Kidney) cells in lines 58 in the revised manuscript.
- Page 2, line 63: Similarly, please provide a description for the CCK-8 reagent.
Answer:Thanks for your comments and suggestion. We have described CCK-8 (Cell Counting Kit-8) in line 75 in the revised manuscript.
- Page 4, line 140: In Figure 1B, the red line represents the average percentage of cell viability for MH-S cells exposed to various concentrations of CoPP (0, 5, 10, 25, 50, and 100 μM) over a duration of 24 hours. The absence of standard deviation plotted in the figure is indeed a valid point of consideration. To enhance the data representation and reinforce the statistical significance, it is recommended to include error bars in future figures, which will add clarity and credibility to the results.
Answer:Thanks for your comments and suggestion. We have repeated three independent experiments, and the result represented error bars in revised Figure 1B.
- In Figure 5A, the choice of colors and line types for the groups makes it difficult to identify and separate them. The authors should improve the presentation of the figure to make it clearer and facilitate the visualization of the distinct groups.
Answer:Thanks for your comments! We have changed color of line in revised Figure 5A to make it clearer and facilitate the visualization of the distinct groups.
- Authors should include a paragraph discussing the limitations of the study. Several constraints were identified, emphasizing the necessity to evaluate CoPP's antiviral potential in clinically relevant animal models, specifically horses or mules. Additionally, a more extensive range of CoPP concentrations should be investigated to determine optimal dosages. Understanding the potential development of viral resistance is crucial to assess the long-term efficacy of CoPP as an antiviral treatment. Moreover, comprehensive assessments of the compound's safety and toxicity in non-infected cells and animals are essential prerequisites to ensure its viability as a safe and effective antiviral therapeutic strategy.
Answer:Thanks for your comments and suggestion. We have re-edited “Discussion” section in lines 304-309, 311-334 in the revised manuscript.
- The authors need for a more comprehensive analysis and comparison of the study results with other research that has evaluated the antiviral activity of CoPP against different species and viruses. The authors could consider discussing and comparing their findings with relevant studies that have investigated the effects of CoPP on other viral infections in various animal models. This comparison would provide a broader context for understanding the potential application and effectiveness of CoPP as an antiviral agent. It would also help identify any similarities or differences in the mechanisms of action observed across different viruses and species. Furthermore, the authors might address any existing gaps or discrepancies between their study results and previous research on CoPP's antiviral properties. By acknowledging these discrepancies, they can provide insights into potential factors influencing the variability of CoPP's antiviral effects and propose future directions for more in-depth investigations. By including a more comprehensive discussion and comparison with related studies, the authors can strengthen the significance of their findings and offer a clearer perspective on the potential implications of CoPP as a therapeutic agent for viral infections.
Answer:Thanks for your comments and suggestion. We have re-edited “Discussion” section in lines 304-309, 311-334 in the revised manuscript.
8) Below is a list of studies that the authors can use to better discuss the work:
8.1) Ma LL, Zhang P, Wang HQ, Li YF, Hu J, Jiang JD, Li YH. heme oxygenase-1 agonist CoPP suppresses influenza virus replication through IRF3-mediated generation of IFN-α/β. Virology. 2019 Feb;528:80-88. doi: 10.1016/j.virol.2018.11.016.
8.2) Toro A, Ruiz MS, Lage-Vickers S, Sanchis P, Sabater A, Pascual G, Seniuk R, Cascardo F, Ledesma-Bazan S, Vilicich F, Vazquez E, Gueron G. A Journey into the Clinical Relevance of Heme Oxygenase 1 for Human Inflammatory Disease and Viral Clearance: Why Does It Matter on the COVID-19 Scene? Antioxidants (Basel). 2022 Jan 29;11(2):276. doi: 10.3390/antiox11020276.
8.3) El Kalamouni C, Frumence E, Bos S, Turpin J, Nativel B, Harrabi W, Wilkinson DA, Meilhac O, Gadea G, Desprès P, Krejbich-Trotot P, Viranaïcken W. Subversion of the Heme Oxygenase-1 Antiviral Activity by Zika Virus. Viruses. 2018 Dec 20;11(1):2. doi: 10.3390/v11010002. PMID: 30577437; PMCID: PMC6356520.
8.4) Zhong M, Wang H, Ma L, Yan H, Wu S, Gu Z, Li Y. DMO-CAP inhibits influenza virus replication by activating heme oxygenase-1-mediated IFN response. Virol J. 2019 Feb 20;16(1):21. doi: 10.1186/s12985-019-1125-9. PMID: 30786886; PMCID: PMC6381609.
8.5) Liu H, Li C, He W, Chen J, Yang G, Chen L, Chang H. Free ISG15 inhibits Pseudorabies virus infection by positively regulating type I IFN signaling. PLoS Pathog. 2022 Oct 31;18(10):e1010921. doi: 10.1371/journal.ppat.1010921. PMID: 36315588; PMCID: PMC9648840.
Answer:Thanks for your comments and suggestion. We have re-edited “Discussion” section in lines 304-309, 311-334 in the revised manuscript.
Round 2
Reviewer 1 Report
Dear Authors,
the paper was revised according to the reviewer's comments still some improvement are needed.
In the attached version I highlight the things that should be revised.
Everywhere EqHV-8 non EqHV8
Simple summary, induced: use the present
Agents: here and elsewhere, I suggest to use substances or molecules
Material and methods
The m&m are still confused. I suggested to separate the experimental phase from the methods employed (PCR and IB). PCR and IB should go under the same headline in two separate sub headlines, and moved at the end of the section. Same for virus titration, i suggest to move it or togheter with PCR or in the headings 2.1, that is called 2.1. Cells, viruses, antibodies, and chemicals. Same for 2.10 and 2.11 that should go as subheadings of Animal experiments.
Heading of the results (and MM): You should not insert the conclusion in the title of the headings, ex. 3.3. CoPP reduces EqHV8 infection at different stages should become Effects of CoPP on EqHV-8 infection . At different stages is unclear. Same for the others.
Author Response
- Everywhere EqHV-8 non EqHV8
Answer: We are very appreciated for your suggestion. We have changed all the “EqHV8” into “EqHV-8” in revised manuscript and Figures.
- Simple summary, induced: use the present
Answer: We are very appreciated for your suggestion. We have changed “induced” to “presented” in line 8 in revised manuscript.
- Agents: here and elsewhere, I suggest to use substances or molecules
Answer: We are very appreciated for your suggestion. We replaced the agents with molecules. in lines 9, 16, 325, 339 in revised manuscript.
- The m&m are still confused. I suggested to separate the experimental phase from the methods employed (PCR and IB). PCR and IB should go under the same headline in two separate sub headlines, and moved at the end of the section. Same for virus titration, i suggest to move it or togheter with PCR or in the headings 2.1, that is called 2.1. Cells, viruses, antibodies, and chemicals. Same for 2.10 and 2.11 that should go as subheadings of Animal experiments.
Answer: We are very appreciated for your suggestion. We have moved PCR and IB to under the same headline in two separate sub headlines, virus titration into the headings 2.1, and 2.10 and 2.11 as subheadings of Animal experiments, and renumbered them.
- Heading of the results (and MM): You should not insert the conclusion in the title of the headings, ex. 3.3. CoPP reduces EqHV8 infection at different stages should become Effects of CoPP on EqHV-8 infection. At different stages is unclear. Same for the others.
Answer: We are very appreciated for your suggestion. We have re-edited heading of the results and M&M in lines 78, 87, 93, 100, 143, 144, 160, 188, 205, 225, 257 in revised manuscript.